# HOXB8 Counteracts MAPK/ERK Oncogenic Signaling in a Chicken Embryo Model of Neoplasia

**DOI:** 10.3390/ijms22168911

**Published:** 2021-08-18

**Authors:** Axelle Wilmerding, Lauranne Bouteille, Lucrezia Rinaldi, Nathalie Caruso, Yacine Graba, Marie-Claire Delfini

**Affiliations:** 1Aix Marseille Université (AMU), Centre National de la Recherche Scientifique (CNRS), Institut de Biologie du Développement de Marseille (IBDM-UMR 7288), 13288 Marseille, France; axelle.wilmerding@univ-amu.fr (A.W.); lauranne.bouteille@univ-amu.fr (L.B.); lrinald1@bidmc.harvard.edu (L.R.); nathalie.caruso@univ-amu.fr (N.C.); 2Beth Israel Deaconess Medical Center, Department of Medicine and the Cancer Center, Division of Hematology, Harvard Initiative of RNA Medicine, Harvard Medical School, Boston, MA 02115, USA

**Keywords:** HOX, MAPK/ERK, cancer, developing spinal cord, transcription, chicken embryo

## Abstract

HOX transcription factors are members of an evolutionarily conserved family of proteins required for the establishment of the anteroposterior body axis during bilaterian development. Although they are often deregulated in cancers, the molecular mechanisms by which they act as oncogenes or tumor suppressor genes are only partially understood. Since the MAPK/ERK signaling pathway is deregulated in most cancers, we aimed at apprehending if and how the Hox proteins interact with ERK oncogenicity. Using an in vivo neoplasia model in the chicken embryo consisting in the overactivation of the ERK1/2 kinases in the trunk neural tube, we analyzed the consequences of the HOXB8 gain of function at the morphological and transcriptional levels. We found that HOXB8 acts as a tumor suppressor, counteracting ERK-induced neoplasia. The HOXB8 tumor suppressor function relies on a large reversion of the oncogenic transcriptome induced by ERK. In addition to showing that the HOXB8 protein controls the transcriptional responsiveness to ERK oncogenic signaling, our study identified new downstream targets of ERK oncogenic activation in an in vivo context that could provide clues for therapeutic strategies.

## 1. Introduction

The HOX family is an evolutionary conserved set of proteins that regulate developmental processes such as anteroposterior body axis patterning, organ morphogenesis and cell fate. As transcription factors, their mainly act by target gene transcriptional activation or repression [1,2]. There are 39 HOX genes in vertebrates, arranged in a contiguous manner in four groups of 9-11 genes, A, B, C and D, located in humans on 4 different chromosomes, at 7p15, 17p21, 12q13, and 2q31, respectively [3]. HOX genes are numbered 1 through 13 so that each HOX has up to four paralogs on each of the four chromosomal loci. Their role in embryonic development during which their spatial and temporal expression is tightly governed is well established [2]. HOX genes remain expressed in adult tissues and participate, among other things, in tissue-specific stem cell differentiation contributing to the maintenance and function of various organs and tissues [4].

Aberrant expression of HOX genes is associated with many human pathologies including cancers [5,6]. The roles of HOX transcription factors as oncogenes or tumor suppressor genes, depending on the cellular context, are well-established and imply modulation of angiogenesis, differentiation, apoptosis, proliferation, epithelial-to-mesenchymal transition and DNA repair [3]. Highlighting the precise context-dependent and versatile role of the HOX proteins, the same HOX protein can act either as a tumor suppressor or as an oncogene [7]. For example, HOXA9 drives leukemia [8], but suppresses metastasis and growth in breast cancer [9]. Similarly, HOXB8 acts as an oncogene in several cancers including colorectal cancer [10], but is a favorable prognostic marker in renal cancer (Human Protein Atlas) [11]. HOXB8 has also been recently shown to positively control the transcriptional expression of the tumor suppressor *LZTS1* [12]. A better understanding of the molecular mechanisms that promote either the oncogenic or tumor suppressor activity of the HOX proteins remains unsolved.

The mitogen-activated protein kinase (MAPK) cascades are critical pathways for human cancer cell survival, dissemination, and resistance to drug therapy [13,14]. In particular, the MAPK/extracellular signal-regulated kinase (ERK1/2) pathway is a signaling node that receives input from numerous stimuli, including internal metabolic stress and DNA damage pathways, as well as through signaling from external growth factors, cell-matrix interactions, and communication from other cells [15]. Through sequential activation of Ras-like GTPase, RAF, MEK, and ERK1/2, the ERK1/2 pathway transduces the signal of growth factors across the cell membrane to control cell cycle progression, proliferation, survival, and differentiation. This pathway is finely regulated by feedback loops and nearly strictly converges towards the phosphorylation and activation of the ERK1 and ERK2 kinases which interact with many cytosolic and nuclear substrates (>300 described) [16,17]. The induction of immediate early genes (IEGs) following exposure to ERK1/2 represents the first major transcriptional program that precedes changes in a variety of cellular responses. The MEK1/2-ERK signaling directly activates IEG promoter-bound transcription factors [18] and activates the transcription of such genes as the Fos, Jun and Myc families of transcription factors, as well as other transcription factors, such as Egr-1 [19].

That overactivation of the RTK/RAS/RAF/MEK/ERK signal is a key oncogenic event is known for more than two decades [20] and has been recently confirmed by pan-cancer genomic analyses, which have indeed shown that it is the signaling pathway with the highest median frequency of alterations (46% of the samples) across all cancer types [21]. The clinical and therapeutic success of RAF and MEK1/2 inhibitors has revolutionized the existing treatment schemes for previously incurable cancers. However, the overall therapeutic efficacies are still largely compromised by side effects and emerging drug resistance mechanisms [16]. It is therefore crucial to improve our understanding of the molecular mechanisms underlying the oncogenic activity of the ERK pathway to develop more effective therapies.

Studies on zebrafish embryos suggested that posterior Hox genes control responsiveness to the RTK/FGF signal (which has ERK as the cellular effector in this tissue) [22] during early development [23]. In addition, recent studies have shown that HOXB7, HOXA3 and HOXC6 act as oncogenes by activating the ERK pathway in pancreatic, colon and glioblastoma cancers, respectively [24,25,26]. To gain insights in the molecular mechanisms by which the HOX proteins control ERK oncogenic pathways, we developed a chicken embryo neoplasia model obtained by ERK1/2 overactivation in the trunk neural tube [27] and investigated the effects of HOX expression in this model. We started the analysis with HOXB8, for which we have recently investigated the function during the development of the spinal cord and identified transcriptional target genes in this structure [12]. We found that HOXB8 largely counteracts ERK oncogenic activity in this neoplasia model. Transcriptomic analysis showed that the HOXB8 tumor suppressor function relies on a large reversion of the oncogenic transcriptome induced by ERK overactivation. HOXB8 effect is independent of ERK phosphorylation and nuclear translocation and does not rely on the transcriptional control of the immediate early genes of the EGR family but might rely on the transcriptional control of the MYC and FOS families. This study also identified new transcriptional downstream targets of ERK controlled by HOXB8 that could provide clues for therapeutic applications. We found that one of the genes most upregulated by ERK overactivation and whose expression is reversed by HOXB8 is the gene encoding CHST15. CHST15 expression is unfavorable in most human cancers, underlining the relevance of this chicken in vivo model to identify new targets relevant in human pathologies.

## 2. Results

### 2.1. HOXB8 Transcription Factor Inhibits Neoplasia Induced by ERK Overactivation in the Chicken Embryo Neural Tube

We have recently analyzed the function and identified target genes of HOXB8 in the developing spinal cord of chicken embryo [12]. In particular, we have demonstrated that the gain of function of HOXB8 in the trunk neural tube of a 2-day-old chicken embryo performed by electroporation of the pCIG-HOXB8 expressing vector (co-expressing nuclear GFP) increases cell death and the amount of the transcript of the gene coding for the tumor suppressor LZTS1 [12]. To test the effect of HOXB8 on ERK oncogenic activity in an in vivo context, we took advantage of a model of ERK-induced neoplasia we developed in the chicken embryo neural tube [27]. The model consists in the expression of a constitutive active form of the kinase MEK1 (MEK1ca) acting upstream of ERK1/2 [28,29], by electroporating the pCIG-MEK1ca expressing vector in the trunk neural tube of 2-day-old chicken embryos. While pCIG-MEK1ca electroporation induces a strong neoplastic phenotype three days post-electroporation [27] (Figure 1A), co-electroporation of pCIG-MEK1ca with pCIG-HOXB8 leads to a milder phenotype (Figure 1A). Immunofluorescence staining on transverse sections with a GFP antibody to stain transfected cells (Figure 1A), co-stained with phalloidin (Appendix A), allowed to highlight that while the neural tube is still morphologically affected (compare to the control/pCIG condition and to the contralateral/non-electroporated part of the neural tube), the extent of neoplasia is much weaker in all the embryos analyzed when MEK1ca is co-transfected with the HOXB8 expressing vector (>10 embryos analyzed for each condition). This morphological analysis suggested that in this context, the HOXB8 gain of function works as a tumor suppressor.

We next decided to investigate the molecular mechanisms by which HOXB8 inhibits ERK1/2 overactivation-induced neoplasia. HOXB8 is endogenously expressed in the trunk neural tube of chicken embryos from E3, controlling neuronal delamination [12]. Since HOXB8 overexpression in the neural tube leads to an increase of apoptosis [12], HOXB8 tumor suppressor effect may be due to cell death induction in MEK1ca expressing cells. We thus performed immunofluorescence staining on transverse sections of chicken embryos one day after electroporation and quantified the cleaved caspase 3 (CASP3) apoptotic marker in the neural tube after transfection of control pCIG, MEK1ca or MEK1ca + HOXB8. We found increased apoptosis in the MEK1ca condition (as already described in [27]) which is further enhanced in the MEK1ca + HOXB8 condition (Figure 1B,C) in agreement with the hypothesis that HOXB8-induced cell death contributes to the reduction of MEK1ca-induced neoplasia. Since the Hox genes have been shown to control the responsiveness of the FGF/ERK signaling in a developmental context in zebrafish [23] we next probed if in addition to increasing cell death in MEK1ca-expressing cells, the HOXB8 gain of function might also counteract MEK1ca-induced oncogenic transcriptome. To test this hypothesis, we performed fluorescence in situ hybridization on transverse sections with probes against target genes of the ERK pathway, the DUSP5, GREB1 and IL17RD genes, whose transcripts are strongly upregulated after MEK1ca expression in the chicken neural tube [27]. We found that when HOXB8 is co-expressed, MEK1ca does not induce the expression of any of these three genes (Figure 1D, Appendix A). We next asked if the suppressing effect of HOXB8 on MEK1ca-induced target genes observed occurs upstream, by modifying ERK1/2 activation, or downstream of ERK1/2 by modifying the transcriptional response to ERK1/2 signaling. We performed immunofluorescence staining on transverse sections with the phospho Thr202/Tyr204 ERK1/2 (pERK) antibody which reflects ERK1/2 kinase activity. We found that HOXB8 co-expression does not change the level of MEK1ca-induced ERK activation, and that pERK staining was still localized in both the cytoplasm and the nucleus, as is the case following MEK1ca induction (Figure 1E,F). Altogether, these results show that HOXB8 functions as a tumor suppressor in this neoplasia context by increasing cell death and preventing MEK1ca-induced target gene activation, acting downstream of ERK phosphorylation and nuclear translocation.

### 2.2. HOXB8 Largely Reverses the ERK-Induced Oncogenic Transcriptome

To identify the global transcriptional changes underlying HOXB8-mediated suppression of MEK1ca-induced neoplasia, we performed RNA-Seq on GFP-sorted positive neural tube cells (Figure 2A). E2 neural tubes were bilaterally electroporated with either the control vector pCIG (control condition), pCIG-MEK1ca [27], pCIG-HOXB8 [12], or pCIG-MEK1ca + pCIG-HOXB8 (Figure 2A). The regions of the neural tube expressing the GFP were dissected 18 h after electroporation and dissociated (18 to 20 embryos are pooled by condition, Figure 2A). GFP-expressing cells were isolated by FACS with the use of a dead cell exclusion (DCE)/discrimination dye (DAPI) to eliminate dying cells. Two independent RNA samples (by condition) were extracted and reverse transcribed. Complementary DNA was amplified using a linear amplification system and used for sequencing libraries. Qualitative analysis of the RNA-Seq data after alignment to the Galgal4 genome assembly (17,108 genes) from the two biological replicates for the four conditions showed a high Pearson correlation coefficient (>0.99) indicative of experimental reproducibility, confirmed by principal component analysis (Appendix A).

Comparison of the differential gene expression (DGE) for FDR1 (False discovery rate ≤ 0.01) between “MEK1ca + HOXB8 versus MEK1ca” (Figure 2B, Appendix A) and “MEK1ca versus pCIG” (Appendix A for FDR1; FDR5 (False Discovery Rate ≤ 0.05) is available in Wilmerding et al., 2021b) identified 759 commonly deregulated genes (Figure 2C). Surprisingly, more than 90% (689 genes) of the commonly deregulated genes were inversely correlated, as illustrated by the expression heatmap (Figure 2D). In other words, among the commonly deregulated genes, most of the genes deregulated in the MEK1ca condition displayed expression levels that tended to go back to their level in the control condition (pCIG) when HOXB8 was co-transfected. Among these genes are DUSP5, IL17RD and GREB1, for which in situ hybridization experiments showed reversion of the MEK1ca-induced phenotype in the presence of HOXB8 (Figure 1D, Appendix A). The RNA-Seq data thus identify a HOXB8-mediated reversion of the MEK1ca-induced transcriptome. Of note, 19 of the 20 genes that were most upregulated by MEK1ca displayed reduced expression (nearly completely for some of them) when HOXB8 was co-expressed (Appendix A).

Using in situ hybridization on tissue sections, we confirmed the RNA-Seq data, including for the genes for which MEK1ca has a modest effect, as illustrated for *LIN28A* (Figure 2E,F): *LIN28A* expression is upregulated by MEK1ca (MEK1ca versus pCIG) [27], and downregulated by HOXB8 alone (HOXB8 versus pCIG) [12] or by the combination of both (MEK1ca + HOXB8 versus pCIG) (Figure 2E). Chicken *LIN28A* is endogenously expressed in E15/E16 embryos [30], the stage at which RNA-Seq was performed. By whole mount in situ hybridization, we found that in the trunk neural tube of chicken embryos, *LIN28A* is expressed in a gradient along the rostro-caudal axis, consistent with regulation by the FGF/ERK signal produced as a gradient from the most caudal part of the embryo (Appendix A) [27]. This posterior gradient was more pronounced at E2, the stage at which we performed electroporations. In situ hybridization on transverse trunk sections showed that MEK1ca upregulated *LIN28A*, while HOXB8 downregulated *LIN28A*, including when co-expressed with MEK1ca (Figure 2F,G).

*LIN28A* encodes highly conserved RNA-binding protein that represses microRNAs including *let-7* and influences mRNA translation, thus regulating self-renewal of embryonic stem cells [31]. *LIN28A* is also important for body growth and metabolism, tissue development and somatic reprogramming. Furthermore, its role as an oncogene has been widely demonstrated [32]. High levels of LIN28A (or LIN28B, the second Lin protein) are associated with many human cancers such as glioblastoma, ovarian, gastric, prostate and breast cancers, as well as pediatric cancers [33]. In mice the ectopic expression of Lin28a is sufficient to induce and/or accelerate tumorigenesis by a let-7-dependent mechanism [32]. In addition to confirm the RNA-Seq data, the study of LIN28A regulation by MEK1ca and HOXB8, together with LIN28A oncogenic activity, suggested that HOXB8 reversion of MEK1ca neoplasia at least partly relies on transcriptional downregulation of LIN28A.

In order to identify other mechanisms that might account for the HOXB8 tumor suppressor function in the MEK1ca context, we performed a gene ontology enrichment analysis using the Enrichr analytical tool [34,35,36] for biological processes (GO Biological Processes 2021), for the upregulated genes in the MEK1ca versus pCIG condition among the 759 commonly deregulated genes (FDR1) (between “MEK1ca + HOXB8 versus MEK1ca” and “MEK1ca versus pCIG”), with inverse correlation (519 genes) (Figure 2H, Appendix A). The hits are coherent with the tumor suppressor role of HOXB8 in this context. Indeed, the genes on this list are part of biological processes such as regulation of cell migration and cell motility, regulation of ERK1/2, intracellular signal transduction, negative regulation of apoptotic processes and extracellular matrix organization (Figure 2H). In conclusion, the RNASeq results, validated by in situ hybridization, and the gene ontology analysis, are consistent with the phenotypic tumor suppressor effect of the HOXB8 gain of function in the MEK1ca-neoplasia-induced context observed morphologically.

### 2.3. Clustering Analysis of RNA-Seq Allows the Identification of New Potential Oncogenes and Tumor Suppressors

To further the molecular mechanisms involved in the HOXB8 tumor suppressor function, and to identify new oncogenes and tumor suppressors, we performed a clustering analysis of the RNA-Seq data starting with the four experimental conditions and using “k-means” on the list of the 2316 genes deregulated upon MEK1ca expression (MEK1ca versus pCIG) (FDR5) [27] (Figure 3 and Appendix A). Twelve is the number of clusters which group together the different dynamics of gene expression in the different conditions, by categories which seemed to us to be the most interesting (i.e., which allow bringing out potential oncogenes, potential tumor suppressors, as well as genes not implicated in the tumor suppressor role of HOXB8 by not being reversed by HOXB8 in the MEK1ca condition).

42.3% (980/2316) of the genes have their expression increased by MEK1ca (MEK1ca versus pCIG) and reversed by HOXB8 in the MEK1ca context (downregulated in MEK1ca + HOXB8 versus MEK1ca) (clusters 1–5, Figure 3A,B, Appendix A). Therefore, this list of genes might contain new putative oncogenes. Among them, cluster 1 (351 genes) regroups genes that are not regulated by HOXB8 alone (Figure 3B). Panther overrepresentation analysis for the biological process with cluster 1 genes (Appendix A) highlights the enrichment of genes involved in cell migration and positive and negative regulation of MAPK/ERK signaling (Appendix A). A deeper analysis of these genes based on the literature revealed that some of them are (1) genes which play a physiological role in maintaining cells in an immature state downstream of the FGF/ERK pathway that are normally expressed in neuromesodermal progenitors (NMP) (including CDX4, CA12, PLD1, PVALB, RHOV, COBL, WNT5B, SPP1, CFC1, HCRTR2, DUSP10, CRHB, ADRA2A, F2RL1, TMEM154, RASGRP1, PRR16, PAX5) when compared to the list of genes specifically enriched in human NMP-like cells [37], (2) genes controlling the FGF/ERK signaling pathway (including DUSP5, DUSP6, DUSP10, DAB2IP, SPRY2 and IL17RD), and (3) genes which are endogenously not expressed in the embryo at that stage and thus do not control rostro-caudal maturation of embryos in physiological conditions (including IL1R1, AQP1 and ARAP3) (Figure 3B and Appendix A) [27].

In the group of putative oncogenes (clusters 1–5), clusters 2 and 4 display the same expression dynamics (although the pattern was more pronounced for cluster 2), i.e., both were downregulated by HOXB8 alone (HOXB8 versus pCIG) and in the MEK1ca context (MEK1ca + HOXB8 versus MEK1ca) (Figure 3B, Appendix A). Panther overrepresentation analysis for the biological process with these genes (clusters 2 + 4, Appendix A) highlighted enrichment of the genes involved in the regulation of translation initiation, including LIN28A (Figure 3B), known precisely for controlling this function [31], and also, among others, of the genes involved in cortical actin cytoskeleton organization (Appendix A). Due to their negative transcriptional regulation by HOXB8 in the physiological context, some of the genes in clusters 2 + 4 may contribute to reverse the MEK1ca-induced oncogenic phenotype. Among these genes is, for example, KSR1 gene (cluster 2, Figure 3B and Appendix A). It encodes a molecular scaffold kinase suppressor of RAS which plays potent roles in promoting RAS-mediated signaling through the RAF/MEK/ERK kinase cascade [38]. Another gene of this group that might be interesting to consider is TMEM132C (cluster 2, Figure 3B). Indeed, the molecular functions of the TMEM132 genes remain poorly understood and under-investigated despite their mutations associated with cancer [39]. Structural analysis of this gene predicted a cellular adhesion function, connecting the extracellular medium with the intracellular actin cytoskeleton [39]. For this gene, we described for the first time its expression in vertebrate embryo (Figure 4A). Its expression during the development of the spinal cord suggests that it might play a physiological function in the differentiation of motor neurons and in the fate and/or migration of neural crest cells of the trunk (Figure 4A). Outside of the neural tube, it is also expressed in the dorsal root ganglia and in the myotome, where it might also play key functions (Figure 4A). In situ hybridization on tissue section (Figure 4C) confirmed the dynamics of *TMEM132C* expression identified from the RNA-Seq data (Figure 4B).

30.4% (704/2316) of the genes (clusters 6–8) behaved oppositely to the genes of clusters 1–5: their expression was downregulated by MEK1ca (MEK1ca versus pCIG) and reversed by HOXB8 (upregulated in “MEK1ca + HOXB8 versus MEK1ca”), and thus might contain new putative tumor suppressors (Figure 3B, Appendix A). Strikingly, panther overrepresentation analysis for the biological process with these genes (clusters 6–8, Appendix A) highlighted enrichment of the genes involved in the negative regulation of cell growth and the regulation of neuron apoptotic processes (Appendix A) in agreement with the tumor suppressor function. Among these genes, some of them were not regulated by HOXB8 alone (HOXB8 versus pCIG) (cluster 6, Figure 3B), such as DBX2 (Appendix A) involved in primary neurogenesis [40]. They also include genes regulated by HOXB8 alone (HOXB8 versus pCIG) (clusters 7–8, Figure 3B), such as *LZTS1* (cluster 7, Figure 3B and Appendix A), a leucine zipper tumor suppressor that suppresses colorectal cancer proliferation through inhibition of the AKT pathway [41] and that we have recently found to control neuronal delamination in the trunk neural tube of chicken embryo [12]. *LZTS1* is transcriptionally activated by HOXB8 and might, as other genes from the same cluster, provide additional mechanisms by which HOXB8 reverses MEK1ca-induced neoplasia. *NELL2* (Figure 3B and Appendix A) (neural EGFL like 2) for example, expressed in the neural tube and involved in neural development [42], has also been described as a tumor suppressor since it is enriched in normal nerve cells compared with nervous system tumors [43] and inhibits cancer cell migration in renal cell carcinoma [44].

### 2.4. Among the Genes Not Reversed by HOXB8 Are Immediate Early Genes EGR1 and EGR4

Clustering analysis of the RNA-Seq data also allowed regrouping together the genes not reversed by HOXB8 (expression not changed between the MEK1ca + HOXB8 versus MEK1ca condition). These genes, grouped in cluster 12, represents only 6% of the genes analyzed (114/2316) (Figure 3B, Appendix A). Panther overrepresentation analysis for the biological process with these genes highlighted enrichment of the genes involved in the regulation of the cell-matrix adhesion, cell differentiation or regulation of transcription by RNApolII (Appendix A). Surprisingly, among the genes of this category were included early growth response genes (EGR1 and EGR4) (Appendix A) known to be important transcriptional regulators and to act as the convergent point between a variety of extracellular stimuli including the RAS/ERK pathway and activation of their target genes [19,45]. Indeed, while they were, as expected, upregulated by MEK1ca (MEK1ca versus pCIG), they were not reversed by HOXB8 (expression not changed between the MEK1ca + HOXB8 versus MEK1ca condition) (Figure 5A and Appendix A). In situ hybridization with the EGR1 probe confirmed the dynamics of expression identified from the RNA-Seq data (Figure 5A,C). Other genes which behave the same are genes coding for ETS transcription factors ETV3 and ETV5 (cluster 12, Appendix A, Figure 3B, Figure 5A, Appendix A), also known to be the downstream target of the FGF8/ERK pathway [46]. These results highlighted the specificity of the mechanisms leading to HOXB8 reversion of MEK1ca-induced neoplasia. Since other immediate early genes are however part of the genes reversed by HOXB8 expression, including c-MYC [19] (cluster 4, Figure 5B, Appendix A), the global transcriptional reversion of the MEK1ca phenotype by HOXB8 could rely on the transcriptional control of these genes by HOXB8 which might work as nodes of the global transcriptional regulation of the other genes. The MYC genes, which consist of three paralogs encoding transcription factors C-MYC, L-MYC and N-MYC, are among the most frequently deregulated driver genes in human cancer [47]. Since our transcriptomic data showed that HOXB8 inhibited the expression of all the three MYC genes in the trunk neural tube [12] (Appendix A), one simple explanation of the global transcriptional reversion of the MEK1ca-induced phenotype by HOXB8 is via MYC transcriptional inhibition, that could be direct or indirect. It fits at least for the LIN28A gene which is controlled at the transcriptional level via MYC [48]. However, we do not rule out that other genes such as FOS (FOSL2 gene is in the cluster 4, Figure 3B, Appendix A) or other immediate early genes also act as relay of MEK1ca targets reversed by HOXB8, or that HOXB8 acts more as a global epigenetic factor which controls transcription in direct competition with ERK or one (or few) of its key target effectors.

### 2.5. CHST15, a Target of HOXB8 Transcriptional Reversion of MEK1ca-Induced Neoplasia, Correlates with Poor Survival in Many Human Cancers

Among the genes of cluster 1 (Figure 3B), the *CHST15* gene caught our attention because it represents one of the genes most upregulated by MEK1ca and reversed by HOXB8 (after IL1R1, CDX1 like, CDX4 and DUSP5 and just before AQP1, Appendix A). carbohydrate sulfotransferase 15 (CHST15) is a specific enzyme that biosynthesizes Chondroitin sulfate E (CS-E) [49], a highly sulfated glycosaminoglycan promoting tumor invasion and metastasis. Although CHST15 has been described in the literature as being oncogenic in some cancers, including in breast cancer [50], its regulation by the ERK pathway or by HOX transcription factors has not been described. During embryonic development, *CHST15* is expressed very specifically in the most caudal region of the chicken embryo [51], where FGF8 signaling and pERK are high [22,27] (Figure 6A). We validated the RNA-Seq results obtained for *CHST15* (Figure 6B) by in situ hybridization on chicken embryo transverse sections with a *CHST15* probe: MEK1ca increased *CHST15* expression only in the absence of HOXB8 (Figure 6C,D). Using data from the Human Protein Atlas (TCGA RNA-Seq data) (courtesy of the Human Protein Atlas, www.proteinatlas.org, 15 May 2021) [11], we highlighted that CHST15, which had already been described as an unfavorable prognostic marker for renal cancer, displayed a high expression correlated with poor outcomes in most human cancers in the survival analysis (15/17 cancer types) (Figure 6E,F and Appendix A). In conclusion, our finding that CHST15, which seems to be unfavorable for most human cancers, is a positive target of ERK in oncogenic conditions, and is transcriptionally reversed by the HOXB8 transcription factor gain of function, underlines the relevance of the chicken MEK1ca/HOXB8 in vivo model to identify new key oncogenes and tumor suppressors, pertinent in human pathologies.

## 3. Discussion

The RAS/RAF/MEK/ERK (MAPK/ERK) pathway is hyperactivated and takes an active part in the malignant transformation in most cancers [52,53,54], including melanoma [55], and neuroblastoma [56] (cancers that have neural tube cells as the embryological origin). Inhibitors of this pathway are used clinically and improve the prognosis of cancer patients for melanomas but are poorly efficient in other cancers [57]. In addition, a major problem with the current therapies is that patients often develop resistance [53]. Although immunotherapy has recently emerged as an effective therapeutic approach [58], ERK1/2 activity inhibition is still considered a prime target for the treatment of most cancers. As a result, much effort is being made to understand the barriers to current treatments and discover new therapeutic strategies to counteract ERK hyper-activity in oncology [53].

In this study, we used the chicken embryo neural tube as a platform to explore novel regulatory/interfering paths for ERK oncogenic activity. The model of neoplasia induced by MEK1ca expression we recently developed [27] is experimentally convenient, economic, respects the 3R [59] and allows apprehending the epistatic relationship between the ERK pathway and the other proteins including transcription factors during oncogenic progression. Furthermore, the results obtained identified putative new oncogenes and tumor suppressor genes, that may be particularly relevant for cancers originating from the embryonic neural tube. The data presented also suggest several non-exclusive mechanisms though which HOXB8 acts as tumor suppressor in the neural tube downstream of oncogenic ERK activation. This includes the increase of cell death, direct or indirect transcriptomic inhibition of immediate early genes including MYC (c-MYC, MYC-L and MYC-N) and FOS (but not EGR1 and EGR4) which control the transcription of a myriad of downstream targets, and the activation of tumor suppressor genes such as *LZTS1*.

One of the transcriptional targets of MYC is *LIN28A* [48] which we found upregulated by MEK1ca and reversed by HOXB8 suggesting that HOXB8 might at least counteract MEK1ca oncogenic activity by negatively regulating LIN28A. The recent results obtained in mice suggest that during caudal bud development, the expression of the *Lin28* genes is controlled by the Hox proteins (Hoxb13 and Hoxc13) [60]. The transcriptional control of the *LIN28* genes by the HOXB8 protein in the chicken neural tube context could thus be a function shared between the HOX proteins and not restricted to this tissue (neural tube) and organism (chicken). We recently highlighted shared or generic HOX functions during the development of the spinal cord in chicken embryo [12], and, more generally, in other biological contexts and species [61,62]. It would thus be interesting to investigate if other HOX proteins behave as HOXB8 in the specific context of MEK1ca neoplasia induction in the embryonic neural tube.

Previous studies also described the HOX genes as tumor suppressors. For example, elevated HOXB9 expression predicts a favorable outcome in colon adenocarcinoma patients [63]. Furthermore, HOXA5 is downregulated in colon cancer, and its re-expression induces loss of the cancer stem cell phenotype, preventing tumor progression and metastasis [64]. Of note, in renal cancer, HOXB8 is prognostic, with high expression depicted as favorable (Human Protein Atlas resource) [11]). Whether these HOX tumor suppressor activities operate by counteracting ERK oncogenic activity already associated with these cancers [65] is, to our knowledge, not known.

Finally, our work opens perspectives to explore novel ERK-interfering strategies with longer-term therapeutic potential. First, it identifies the means to globally block transcriptional changes induced by ERK signaling, downstream ERK phosphorylation and translocation in the nucleus. This defines a biological context and a unique tool to explore ERK interfering approaches in the nucleus. Second, it identifies the genes (IL1R1, CDX1 like, CDX4, DUSP5, and AQP1) (Appendix A) strongly activated by MEK1ca and nearly fully reversed by HOXB8 which may be used as readouts to screen in an in vivo vertebrate context for new molecules capable of counteracting ERK activity downstream of pERK nuclear translocation. To render the MEK1ca neoplasia model more convenient and independent of the electroporation step, chicken/quail transgenic animals allowing for in ovo doxycycline induction of MEK1ca could be generated.

## 4. Materials and Methods

### 4.1. Ethics Statement

The chicken embryos used in this study were all in early stages of embryonic development (between E2 and E5), Therefore, no approval from the Institutional Animal Care and Use Committee was necessary to perform the embryo experiments.

### 4.2. Chicken Embryos

Fertilized chicken eggs were obtained from EARL les Bruyeres (Dangers, France) and incubated horizontally at 38 °C in a humidified incubator. The embryos were staged according to the developmental table of Hamburger and Hamilton (HH) [66] or according to the days of incubation (E).

### 4.3. In Ovo Electroporation and Plasmids

Neural tube *in ovo* electroporation was performed around HH11. The eggs were windowed, and the DNA solution was injected in the neural tube lumen. Needle L-shaped platinum electrodes (CUY613P5) were placed on both sides of the embryo at trunk level (5 mm apart), with the cathode always at its right. Five 50 ms pulses of 25 volts were given unilaterally (or bilaterally for the RNA-Seq experiments) at 50 ms intervals with NEPA21 electroporator (Nepagene).

The plasmids used for the gain-of-function experiments co-express a cytoplasmic or nuclear GFP (pCAGGS and pCIG, respectively, used alone as controls) and the coding sequence (CDS) of the gene of interest. The vectors used were: pCIG-MEK1ca [28], pCIG-HOXB8 (gifted by Doctor Olivier Pourquié). In the trunk neural tube of the chicken embryo at that stage, co-transfection of two plasmids is nearly total [12]. In this study, both plasmids carried the same reporter (GFP). We assumed that all the GFP-expressing cells in the MEK1ca + HOXB8 condition co-expressed MEK1ca and HOXB8.

All the plasmids used for electroporation were purified using a Nucleobond Xtra Midi kit (Macherey-Nagel). The final concentration of DNA delivered by an embryo for electroporation was 2 µg/µL: pCIG (2 µg/µL), pCIG-MEK1ca (1 µg/µL + pCIG 1 µg/µL) and pCIG-HOXB8 (1 µg/µL + pCIG 1 µg/µL).

### 4.4. Tissue Section

The embryos were fixed in 4% buffered formaldehyde in PBS and then treated with a sucrose gradient (15% and 30% in PBS), embedded in the OCT medium and stored at −80 °C. The embryos were sectioned into 16 µm sections with a Leica cryostat and the slides were conserved at −80 °C or directly used for FISH and/or immunofluorescence.

### 4.5. Immunofluorescence

The slides were rehydrated in PBS and then blocked with 10% goat serum, 3% BSA, 0.4% Triton X-100 in PBS for one hour. The primary antibodies were incubated over-night diluted in the same solution at 4 °C. The following primary antibodies were used in this study: chicken anti-GFP 1:1000 (1020 AVES), rabbit anti-phospho-p44/42 MAPK (ERK1/2) (Thr202/Tyr204) antibody #9101 (Cell Signaling Technology, Danvers, MA, USA), rabbit anti-SOX2 1:500 (AB5603 Merck Millipore), mouse anti-Tuj1 1:500 (801202 Biolegend, California), rat anti-pH3 1: 250 (S28, Abcam ab10543), rabbit anti-caspase 3 1:500 (Asp175, CST 9661). The secondary antibodies used were as follows: anti-chicken, anti-rabbit, anti-mouse or anti-rat with fluorochromes (488, 568 or 647) at 1:500 (Alexa Fluor, Abcam, Cambridge). They were incubated for one hour in the blocking solution containing Hoechst (1:1000). F-actin staining was performed using phalloidin AlexaFluor568 (1:70) (Thermo Fisher, Waltham, MA, USA). The slides were washed, mounted (Thermo Scientific Shandon Immu-Mount) and imaged with a ZI Zeiss microscope equipped with an ApoTome or a confocal LSM 780.

### 4.6. In Situ Hybridization

The RNA probes used for in situ hybridization were as follows: DUSP5, IL17RD, LIN28A, EGR1 CHST15 (Wilmerding et al., 2021b), TMEM132C (fw: CTGCCTTGAAATGGCCGGT; rev: AGGTGTCTGCACCAGATCGT), GREB1 (fw: TATGCTGGACAGCTCAAGACA, rev: TTGCGCCCATTATCATCTGGA). The probes were produced from the PCR product amplified from neural tube cDNA from the E3 chicken embryo (WT or transfected by MEK1ca). All the forward primers contained the T7 RNA polymerase promotor sequence: TAATACGACTCACTATAGGGC.

### 4.7. Fluorescence In Situ Hybridization on Tissue Sections

The slides were treated with 10 µg/mL proteinase K (3 min at 37 °C) in a solution of 50 mM TrisHCl pH 7.5, then in 0.1 M triethanolamine and 0.25% acetic anhydride. They were pre-incubated with a hybridization buffer (50% formamide, 5× SSC, 5× Denhardts, 250 µg/mL yeast tRNA and 500 µg/mL herring sperm DNA) for 3h at room temperature and incubated in the same buffer with DIG-labelled RNA probes over-night at 55 °C in a wet chamber. The slides were then washed twice with 0.2× SSC for 30 min at 65 °C. After 5 min in the TNT buffer (100 mM Tris pH 7.5; 150 mM NaCl; 0.1% Tween-20), they were blocked for 1 h in a buffer containing 1× TNT, 1% blocking reagent (Roche) and 10% goat serum, then incubated in the same buffer for 3 h with anti-DIG-POD antibodies (1:500, Roche) and revealed using a TSA-Plus Cyanin-3 kit (Perkin Elmer, Germany).

### 4.8. Whole Mount In Situ Hybridization

The embryos were fixed for 2 h at RT in 4% formaldehyde in PBS. The embryos were dehydrated with sequential washes in 50% ethanol/PBS+ 0.1% Tween20 and 100% ethanol and conserved at −20 °C. The embryos were bleached for 45 min in 80% ethanol + 20% of 30% H2O2 and then rehydrated. They were treated with 10 µg/mL proteinase K at RT and refixed with 4% formaldehyde, 0.2% glutaraldehyde. After 1 h of blocking in a hybridization buffer (50% formamide, 5x SSC, 50 μg/mL heparin, 50 μg/mL yeast tRNA, 1% SDS), hybridization with DIG-labelled RNA probes was performed at 68 °C overnight. The next day, the embryos were washed (three times 30 min each) in a hybridization buffer and once in TBS (25 mM Tris, 150 mM NaCl, 2 mM KCl, pH 7.4) + 0.1% Tween 20. They were incubated for 1 h at RT in a blocking buffer (20% blocking reagent + 20% goat serum) and then overnight with an anti-DIG-AP antibody (1:2000, Roche) in the blocking buffer. After three washes (1 h) in TBS + 0.1% Tween 20, the embryos were equilibrated (two times 10 min each) in an NTMT buffer (100 mM NaCl, 100 mM TrisHCl pH 9.5, 50 mM MgCl_2_, 2%Tween 20) and incubated in NBT/BCIP (Promega) at RT in the dark until color development. Pictures of the whole embryos were made using a BinoFluo MZFLIII and a color camera.

### 4.9. RNA-Seq Analysis

Electroporations were carried out as described in a previous section but with 5 bilateral pulses, with the protocol described in [12]. Results for pCIG (2 µg/µL), pCIG-MEK1ca (1 µg/µL + pCIG 1 µg/µL) and pCIG-HOXB8 (1 µg/µL + pCIG 1 µg/µL) were published in [12,27], and compared with the pCIG-MEK1ca + pCIG-HOXB8 (1 µg/µL of each) condition. Part of the neural tube expressing the GFP were dissected 18 h after electroporation and dissociated (Trypsin-EDTA 0.25%). GFP-expressing cells were isolated by FACS with the use of a dead cell exclusion (DCE)/discrimination dye (DAPI) to eliminate dying cells [12]. RNA was extracted (RNeasy Mini Kit) and reverse transcribed and cDNA was amplified using a linear amplification system and used for sequencing library building (GATC): random primed cDNA library, purification of poly-A containing mRNA molecules, mRNA fragmentation, random primed cDNA synthesis, adapter ligation and adapter specific PCR amplification, Illumina technology, 50,000,000 paired end reads with the 2 × 50 bp read length. Bioinformatics analyses were performed using the galgal4.0 chicken genome. Qualitative analysis of the RNA-Seq data from the four biological replicates showed a high Pearson correlation coefficient (>0.99) indicative of experimental reproducibility (Appendix A). The RNA-seq data discussed in this publication have been deposited in NCBI’s Gene Expression Omnibus [67] and are accessible through GEO Series accession numbers GSE162665, GSE182072 and GSE182076.

### 4.10. Quantifications and Statistical Significance

The number of embryos and sections analyzed is indicated in the figure legends. A minimum of three embryos and six sections per embryo were used for quantifications. All the quantifications were performed using the cell counter tool of the Fiji software. The results were analyzed and plotted using the Prism 8 software (GraphPad software). Statistical analyses were performed using a two-tailed Mann-Whitney test and considered significant whith *p*-value < 0.05. All *p*-values are indicated in the graphs. The error bars represent the standard deviation (SD).

## Figures and Tables

**Figure 1 ijms-22-08911-f001:**
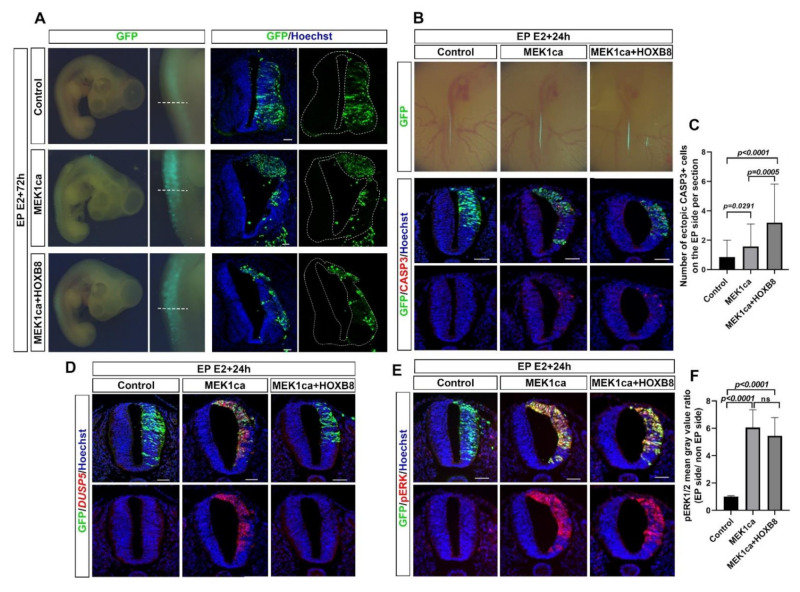
HOXB8 suppresses MEK1ca-induced neoplasia in the trunk neural tube. (**A**) Dorsal view of electroporated embryos three days post-electroporation with the control pCIG-, MEK1ca-, or MEK1ca + HOXB8-expressing vectors. Transfected cells are identified by GFP and immunofluorescence staining on the corresponding transverse sections (white dotted lines) of the chicken embryo with the anti-GFP antibody three days post-electroporation with the pCIG-, MEK1ca-, or MEK1ca + HOXB8- expressing vectors. (**B**) Top: Dorsal view of electroporated embryos one day post-electroporation with the control pCIG-, MEK1ca-, or MEK1ca + HOXB8-expressing vectors. Transfected cells are identified by GFP. Bottom: Immunofluorescence staining on transverse sections of chicken embryo with anti-GFP and anti-cleaved caspase 3 (CASP3) antibodies. (**C**) Number of ectopic CASP3+ cells on the electroporated side one day after electroporation in the three conditions (*n* = 3 animals/18 sections). The quantifications show a significant increase of Casp3+ cells in the neural tube after MEK1ca and MEK1ca + HOXB8 conditions compared to control, with more Casp3+ cells in the MEK1ca + HOXB8 condition than in MEK1ca alone (two-tailed Mann–Whitney test, error bars represent the SD). (**D**) Fluorescence in situ hybridization with the DUSP5 probe and immunofluorescence staining with the anti-GFP antibody on transverse trunk sections of chicken embryo one day post-electroporation in the three conditions. (**E**) Immunofluorescence staining on transverse sections with anti-pERK1/2 and anti-GFP antibodies one day post-electroporation with the pCIG-, MEK1ca- or MEK1ca + HOXB8-expressing vectors. (**F**) one day after electroporation in the three conditions: pERK1/2 mean gray value ratio (electroporated/contralateral side) (*n* = 3 animals/18 sections). The quantifications show no significant difference between the MEK1ca and MEK1ca + HOXB8 conditions (two-tailed Mann–Whitney test, error bars represent the SD). Blue is Hoechst staining. Scale bar: 50 µm.

**Figure 2 ijms-22-08911-f002:**
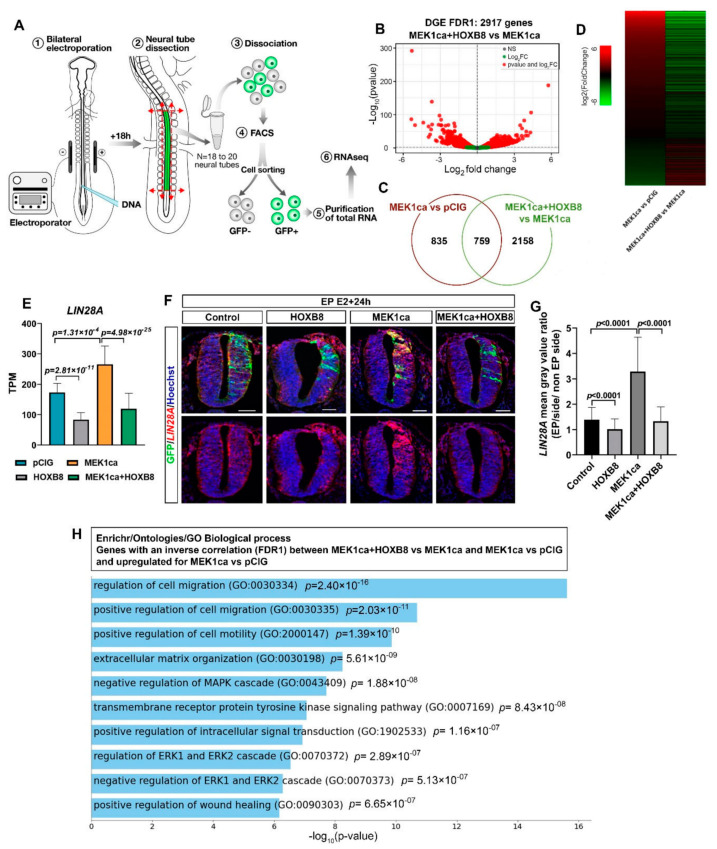
Transcriptomic analysis of MEK1ca-induced neoplasia suppression by HOXB8. (**A**) Experimental design: 18 h after bi-lateral electroporation of the trunk neural tube at stage HH12, the electroporated region of the neural tube was dissected (18–20 embryos per condition in duplicate) and the GFP-positive cells were sorted by FACS. (**B**) Volcano plot of differential gene expression (DGE, FDR1) for the “MEK1ca + HOXB8 versus MEK1ca” conditions (2917 genes). (**C**) Venn diagram showing gene overlaps between the “MEK1ca versus pCIG” and the “MEK1ca + HOXB8 versus MEK1ca” conditions (FDR1). (**D**) Heat map showing the expression of the 759 genes upregulated by MEK1ca and reversed by HOXB8 in “MEK1ca versus pCIG” (line 1) and “MEK1ca + HOXB8 versus MEK1ca” (line 2), highlighting global reversion of MEK1ca-deregulated genes by HOXB8. (**E**) Mean expression of LIN28A TPM (transcripts per kilobase million) obtained for the two replicates of the control- (pCIG1, pCIG2), HOXB8- (HOXB8-1 and HOXB8-2), MEK1ca- (MEK1ca-1 and MEK1ca-2) and MEK1ca + HoxB8 (MEK1ca + HOXB8-1 and MEK1ca + HOXB8-2)-expressing samples, with the corresponding p-value according to the DGE (FDR1) of each pair. (**F**) Fluorescence in situ hybridization with the LIN28A probe and immunofluorescence staining with the anti-GFP antibody on transverse trunk sections of chicken embryos one day post-electroporation in the pCIG, HOXB8, MEK1ca and MEK1ca + HOXB8 conditions. Blue is Hoechst staining. Scale bar: 50 µm. (**G**). LIN28A expression ratio (electroporated/contralateral side) one day after electroporation in the four conditions (*n* = 3 animals/6 sections). The quantifications show that HOXB8 reverses MEK1ca-induced expression of LIN28A (two-tailed Mann–Whitney test, error bars represent the SD). (**H**) Gene ontology enrichment analysis using the Enrichr analytical tool for biological processes (GO Biological Processes 2021), for the upregulated genes in the “MEK1ca versus pCIG” condition among the 759 commonly deregulated genes (FDR1) (between “MEK1ca + HOXB8 versus MEK1ca” and “MEK1ca versus pCIG”), with inverse correlation (519 genes).

**Figure 3 ijms-22-08911-f003:**
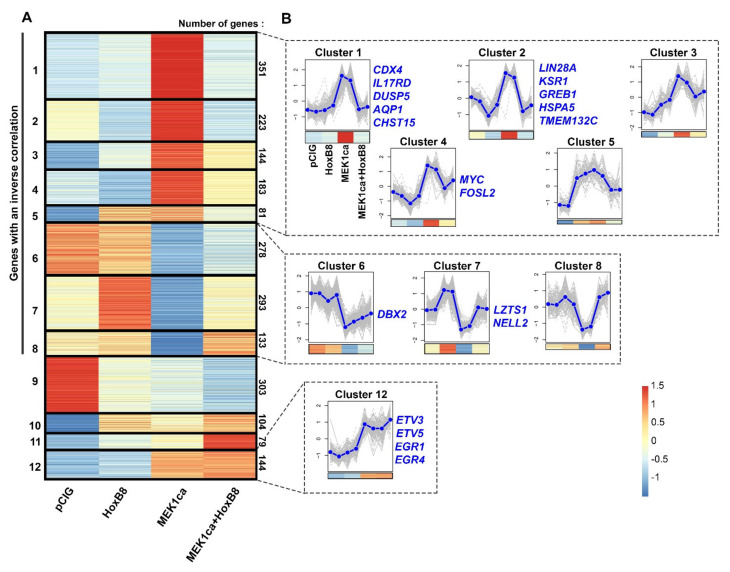
Clustering of genes deregulated by MEK1ca. (**A**) Heat map (*z*-scores derived from TPM values) of the 2316 genes deregulated in the “MEK1ca versus PCIG” condition (FDR5) clustered using *k*-means with *k* = 12 for the pCIG, HOXB8, MEK1ca and MEK1ca + HOXB8 conditions. 1686 genes (72.8%, clusters 1–8) display an inverse correlation between the “MEK1ca versus pCIG” and “MEK1ca + HOXB8 versus MEK1ca” conditions. Only 144 genes (6.2%, cluster 12) behave the same in the MEK1ca and MEK1ca + HOXB8 conditions (i.e., are not reversed by HOXB8). (**B**) Plots of clusters 1–8 and 12 with two replicates of each condition (control (pCIG1, pCIG2), HOXB8 (HOXB8-1 and HOXB8-2), MEK1ca (MEK1ca-1 and MEK1ca-2) and MEK1ca + HOXB8 (MEK1ca + HOXB8-1 and MEK1ca + HOXB8-2)), with the mean (in blue) and with the corresponding heat map in the bottom. Some representative genes of the clusters are on the right side of the corresponding cluster.

**Figure 4 ijms-22-08911-f004:**
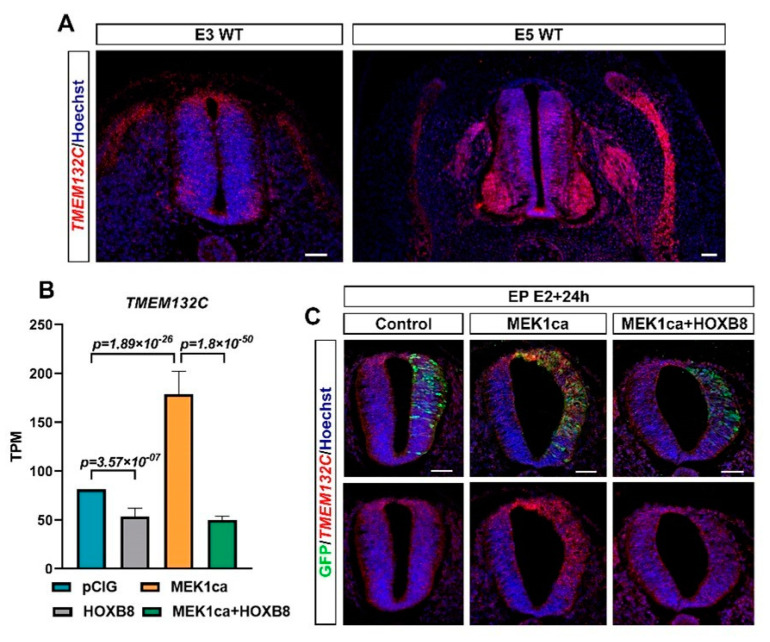
TMEM132C regulation by MEK1ca and HOXB8. (**A**) Fluorescence in situ hybridization with the TMEM132C probe on transverse trunk sections of wild-type chicken embryos at E3 and E5. (**B**) Mean expression of TMEM132C in TPM (transcripts per kilobase million) obtained for the two replicates of the control- (pCIG1, pCIG2), HOXB8- (HOXB8-1 and HOXB8-2), MEK1ca- (MEK1ca-1 and MEK1ca-2) and MEK1ca + HOXB8 (MEK1ca + HOXB8-1 and MEK1ca + HOXB8-2)-expressing samples, with the corresponding p-value according to the DGE of each pair. (**C**) Fluorescence in situ hybridization with the TMEM132C probe and immunofluorescence staining with the anti-GFP antibody on transverse trunk sections of chicken embryos one day post-electroporation in the pCIG, HOXB8, MEK1ca and MEK1ca + HOXB8 conditions.

**Figure 5 ijms-22-08911-f005:**
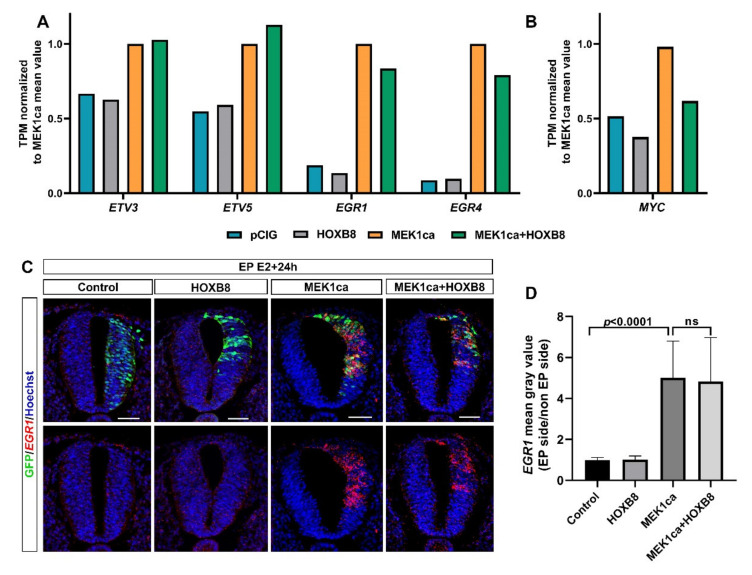
The MEK1ca-induced ETV and EGR gene expression is not reversed by HOXB8. (**A**,**B**) Mean expression in TPM (transcripts per kilobase million) normalized to the MEK1ca value for pCIG, HOXB8 and MEK1ca + HOXB8 (mean of the two replicates of each condition) for the ETV3, ETV5, EGR1 and EGR4 genes (**A**) and the MYC gene (**B**). (**C**) Fluorescence in situ hybridization with the EGR1 probe and immunofluorescence staining with the anti-GFP antibody on transverse trunk sections of chicken embryos one day post-electroporation in the pCIG, HOXB8, MEK1ca and MEK1ca + HOXB8 conditions. Blue is Hoechst staining. Scale bar: 50 µm. (**D**) EGR1 expression ratio (electroporated/contralateral side) one day after electroporation in the four conditions (*n* = 3 animals/18 sections). The quantifications show that HOXB8 does not reverse MEK1ca-induced EGR1 expression (two-tailed Mann–Whitney test, error bars represent the SD).

**Figure 6 ijms-22-08911-f006:**
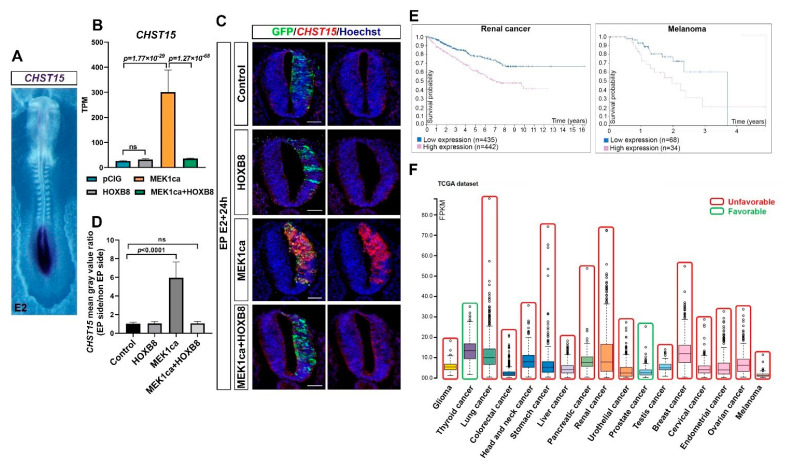
CHST15 upregulated by MEK1ca and reversed by HOXB8 is unfavorable for many cancers. (**A**) Dorsal view of a 2-day-old chicken embryo after whole mount in situ hybridization with a CHST15 probe highlighting its expression only in the most caudal part of the embryo. (**B**) Mean expression of CHST15 in TPM (transcripts per kilobase million) obtained for the two replicates of the control- (pCIG1, pCIG2), HOXB8- (HOXB8-1 and HOXB8-2), MEK1ca- (MEK1ca-1 and MEK1ca-2) and MEK1ca + HOXB8 (MEK1ca + HOXB8-1 and MEK1ca + HOXB8-2)-expressing samples, with the corresponding p-value according to the DGE (FDR5) of each pair. (**C**) Fluorescence in situ hybridization with a CHST15 probe and immuno-fluorescence staining with the anti-GFP antibody on transverse trunk sections of chicken embryos one day post-electroporation in the pCIG, HOXB8, MEK1ca and MEK1ca + HOXB8 conditions. Blue is Hoechst staining. Scale bar: 50 µm. (**D**) CHST15 staining mean gray value ratio (electroporated/contralateral side) one day after electroporation in the four conditions (*n* = 3 animals/18 sections) (two-tailed Mann–Whitney test, error bars represent the SD). (**E**) Survival analysis data from the Human Protein Atlas (courtesy of the Human Protein Atlas, www.proteinatlas.org, 15 May 2021) highlighting that CHST15 displays high expression correlated with poor outcomes in renal and melanoma human cancers. (**F**) RNA expression overview (RNA-Seq data in 17 cancer types as the median FPKM (fragment per kilobase of exon per million reads) generated by The Cancer Genome Atlas (TCGA) (Courtesy of Human Protein Atlas, www.proteinatlas.org, 15 May 2021)), to which we added a color code (red—unfavorable, green—favorable) to highlight that most cancers (15/17) have high expression correlated with poor outcomes in the survival analysis data (Human Protein Atlas).

## Data Availability

The RNA-seq data discussed in this publication have been deposited in NCBI’s Gene Ex-pression Omnibus [67] and are accessible through GEO Series accession numbers GSE162665, GSE182072 and GSE182076. https://www.ncbi.nlm.nih.gov/geo/query/acc.cgi?acc=GSE162665. https://www.ncbi.nlm.nih.gov/geo/query/acc.cgi?acc=GSE182072. https://www.ncbi.nlm.nih.gov/geo/query/acc.cgi?acc=GSE182076 accessed on 10 June 2021.

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
