# Peer review of "HOXB8 Counteracts MAPK/ERK Oncogenic Signaling in a Chicken Embryo Model of Neoplasia"

_ijms, 2021, doi:10.3390/ijms22168911_

Round 1

Reviewer 1 Report

Wilmerding and coworkers, in the manuscript titled “HOXB8 counteracts MAPK/ERK Oncogenic Signaling in a chicken embryo model of Neoplasia,” identified HOXB8 as a tumor suppressor counteracting ERK-induced neoplasia. This study is an exciting direction to study Hoxb8 and its targets for therapeutic intervention for countering cancer progression, which manifests ERK-induced neoplasia. The manuscript contains many exciting experiments and data, but the authors took a casual way to data interpretation and lack experiments to back many important statements. Genome-wide data lacks in-depth analysis, and many genes are cherry-picked to substantiate claims. Hence, the manuscript needs substantial revision before publication.

Major comments 

  1. Interpretation of various clusters of genes showing Differential gene expression needs more careful interpretation. e.g., Line 287, page number 8, “…. increased by MEK1ca and reversed by HOXB8 (clusters 1-5)”. Upon carefully looking into data, interpretation of Cluster 1-5 needs to be more nuanced. For example, Cluster2, where co-expression of MEK1ca+HOXB8 downregulates gene expression compared to pCIG(wt) similarly in Cluster 3, 4, 5 upregulated compared to pCIG. So, if authors consider pCIG as wt, then gene expression needs to be compared to pCIG as a baseline and interpreted accordingly.

  1. Authors assigned various functional roles to multiple clusters, such as cluster 1 genes have a physiological role in maintaining cells in an immature state (Line number 289, page number 8). The authors are not clear about the methodology followed to come to this conclusion. Based on my interpretation, authors are cherry-picking genes and assigning roles to clusters based on the known function of these cherry-picked genes. Genes in each cluster need to be analyzed for functional enrichment using an appropriate tool for enriched GO terms or KEGG pathways. 

  1. The authors claim that HOXB8 transcriptionally regulates MYC and CHST15 genes. There is no direct evidence given to substantiate these claims. Just change in expression levels of these genes cannot be sufficient evidence to support this claim. These changes in expression level could be a direct or indirect effect of HOXB8 on regulating these genes. The authors should identify a regulatory region near these genes which could potentiate regulatory influence through HOXB8 to make claims about transcriptional regulation. 

Minor Comments 

  1. Page 3, line number 101- “The model consists in the expression of a constitutively active form of the kinase MEK1”.. Needs to rewrite this sentence for clarity. 
  2. Page 3, line number 103- “Transfection is obtained by electroporation in the trunk neural tube of 2-days old chicken embryos”. Needs to rewrite this sentence for clarity and Scientific correctness of terminology used. 
  3. Page 4, Line number 151- change “weprobed” to we probed
  4. Page 5, Line number 180 - “cDNAs were amplified using a linear amplification system and used for sequencing library building.” Needs to rewrite this sentence for clarity.
  5. Page 4, Line number 183- change “>0,99” to >0.99
  6. Page 8, Line number 287- change “42,3” to 42.3 

  1. Page 9, line number 330- change “30,4” to 30.4

Figures 

  1. Fig 2 B- Differential expressed genes should be shown in distinct color dots.
  2. Fig 2 H-Separate text from plots to clarity and remove 2018 from the title.
  3. Fig2 H – What values are plotted on X-axis? Values and labels need to be added. 
  4. Fig 2 -Legend -Mention what values are shown in the Heatmap
  5. Fig 3-Legend – Elaborate on what values are used for making Heatmap. Is it z-scores derived from FPKM/TPM values? 

Author Response

Response to Reviewer 1 Comments

Wilmerding and coworkers, in the manuscript titled “HOXB8 counteracts MAPK/ERK Oncogenic Signaling in a chicken embryo model of Neoplasia,” identified HOXB8 as a tumor suppressor counteracting ERK-induced neoplasia. This study is an exciting direction to study Hoxb8 and its targets for therapeutic intervention for countering cancer progression, which manifests ERK-induced neoplasia. The manuscript contains many exciting experiments and data, but the authors took a casual way to data interpretation and lack experiments to back many important statements. Genome-wide data lacks in-depth analysis, and many genes are cherry-picked to substantiate claims. Hence, the manuscript needs substantial revision before publication.

Major comments 

  1. Interpretation of various clusters of genes showing Differential gene expression needs more careful interpretation. e.g., Line 287, page number 8, “…. increased by MEK1ca and reversed by HOXB8 (clusters 1-5)”. Upon carefully looking into data, interpretation of Cluster 1-5 needs to be more nuanced. For example, Cluster2, where co-expression of MEK1ca+HOXB8 downregulates gene expression compared to pCIG(wt) similarly in Cluster 3, 4, 5 upregulated compared to pCIG. So, if authors consider pCIG as wt, then gene expression needs to be compared to pCIG as a baseline and interpreted accordingly.

We thank reviewer 1 for his comment that made us realize that we were not precise enough with our description of the bioinformatic data. The revised version of the manuscript takes into account this remarque. We added in particular in the text that for clusters 1-5, the reversion of HOXB8 that we consider, is in MEK1ca context (MEK1ca+HOXB8 versus MEK1ca condition) and not by comparing with pCIG. More generally, we have clarified this part in the revised manuscript. We hope that reviewer will better appreciate the way we have presented our results and interpreted them.

  1. Authors assigned various functional roles to multiple clusters, such as cluster 1 genes have a physiological role in maintaining cells in an immature state (Line number 289, page number 8). The authors are not clear about the methodology followed to come to this conclusion. Based on my interpretation, authors are cherry-picking genes and assigning roles to clusters based on the known function of these cherry-picked genes. Genes in each cluster need to be analyzed for functional enrichment using an appropriate tool for enriched GO terms or KEGG pathways. 

We thank reviewer 1 for his comment that made us realize that we were not enough clear and had not sufficiently nuanced our claims in this part of the manuscript. We have reworked this part of the results in the revised version by adding panther overrepresentation analysis for biological process for the different clusters described and by being more precise in the description of the methodology and in the analysis of the results.

3. The authors claim that HOXB8 transcriptionally regulates MYC and CHST15 genes. There is no direct evidence given to substantiate these claims. Just change in expression levels of these genes cannot be sufficient evidence to support this claim. These changes in expression level could be a direct or indirect effect of HOXB8 on regulating these genes. The authors should identify a regulatory region near these genes which could potentiate regulatory influence through HOXB8 to make claims about transcriptional regulation. 

We agree with the reviewer 1 that our experiments were not appropriate to determine whether HOXB8 regulate the expression of the deregulated genes identified in this study directly or indirectly. In the revised version, we added that the regulation of these genes by HOXB8 might by direct or indirect.

Identifying HOXB8 regulatory regions near these genes might be the focus of another study.

Minor Comments 

  1. Page 3, line number 101- “The model consists in the expression of a constitutively active form of the kinase MEK1”.. Needs to rewrite this sentence for clarity. 
  2. Page 3, line number 103- “Transfection is obtained by electroporation in the trunk neural tube of 2-days old chicken embryos”. Needs to rewrite this sentence for clarity and Scientific correctness of terminology used. 
  3. Page 4, Line number 151- change “weprobed” to we probed
  4. Page 5, Line number 180 - “cDNAs were amplified using a linear amplification system and used for sequencing library building.” Needs to rewrite this sentence for clarity.
  5. Page 4, Line number 183- change “>0,99” to >0.99
  6. Page 8, Line number 287- change “42,3” to 42.3 

     7. Page 9, line number 330- change “30,4” to 30.4

We addressed all the minor points of reviewer 1 in the revised version of the manuscript and thank him to have pointed them.

Figures 

  1. Fig 2 B- Differential expressed genes should be shown in distinct color dots.

Thank you for this comment, we have changed the volcano plot.

     2. Fig 2 H-Separate text from plots to clarity and remove 2018 from the title.

We changed figure 2H for more clarity.

    3. Fig2 H – What values are plotted on X-axis? Values and labels need to be added. 

The bar graph are ranked by value.

     4. Fig 2 -Legend -Mention what values are shown in the Heatmap

It is the log2FoldChange as indicated next to the Heatmap.

   5.  Fig 3-Legend – Elaborate on what values are used for making Heatmap. Is it z-scores derived from FPKM/TPM values?

It is indeed the z-scores derived from the TPM values, we have added it to the text legend.

Reviewer 2 Report

The manuscript by Wilmerding et al describe the role of HOXB8 as a tumor suppressor in a Chicken Embryo Model of Neoplasia by counteracting ERK signaling. Although the data is convincing in general, some aspects should be further investigated to strengthen the study.

Major comments

  1. The primary focus of the manuscript is elucidating the tumor suppressor role of HOXB8, however the authors do not mention the rationale for selecting HOXB8 among other HOX proteins. How did they narrow down to look at the role of HOXB8 in their model?
  2. It is not clear what the authors are pointing out in Fig 1 A. Fig A and B can be combined as in Fig 1 C. Images in Fig 1 B and subsequent figures, show less transfected cells in the MEK1ca+HOXB8 condition
  3. Also, both MEK1ca and HOXB8 are labelled with GFP and it’s not clear how much of MEK1ca and HOXB8 is expressed in the MEK1ca+HOXB8 condition.
  4. In Fig 1 C image, looks like MEK1ca shows more CASP3 (red) staining than MEK1ca+HOXB8 condition and clearly does not co relate with the quantification in Fig 1D. You can do a western blot for casp3 for the different conditions. Also, does inhibition of CASP3 in the MEK1ca+HOXB8 condition revert the tumor suppressor effect of HOXB8?
  5. The authors also do not show a pCIG-HOXB8 vector alone condition in Fig 1, although they mention HOXB8 is endogenously expressed in the trunk neural tube.
  6. Fig 2D legend for the heat map mentions lines 1 2 and 3 and the Fig shows only 2 conditions and is confusing. I would suggest including genes upregulated by MEK1ca alone

Minor comments

  1. Lines 287 and 330- correct all percentages to 42.3 and 30.4 instead of 42,3 and 30,4
  2. Needs to be re written. Many errors, for example line 300- Among these genes is as example…..line 267- To further the analysis of the RNA-seq to identify molecular mechanisms implicated…line 453- Suggest several non-exclusive mechanisms though which HOXB8 acts as tumor….
  3. Include quantification for all the immunofluorescent images
  4. Fig 4A mentions WT chicken embryo, what is the context of using the term wild type here?
  5. Fig 3 B, mention the genes in cluster 3, 5 and 8?
  6. Line 540 and 551- Italicize
  7. Line 554- 45mn.. 45 min?
  8. Line 555- …They were treated X minutes with proteinase K 10μg/ml at RT. X minutes?
  9. Keep units uniform.. either h or hour, min or minutes
  10. Supplementary fig 6 – Is this image with MEK1ca alone?
  11. Final concentration of DNA delivered to embryo for electroporation is mentioned as between 1 to 2μg/μl. Mention the exact concentrations and volume injected in the neural tube lumen

Author Response

Response to Reviewer 2 Comments

The manuscript by Wilmerding et al describe the role of HOXB8 as a tumor suppressor in a Chicken Embryo Model of Neoplasia by counteracting ERK signaling. Although the data is convincing in general, some aspects should be further investigated to strengthen the study.

Major comments

  1. The primary focus of the manuscript is elucidating the tumor suppressor role of HOXB8, however the authors do not mention the rationale for selecting HOXB8 among other HOX proteins. How did they narrow down to look at the role of HOXB8 in their model?

We thank reviewer 2 for his suggestion to more precisely introduce why we have selected HOXB8 among other HOX proteins. We added the sentence “We started the analysis with HOXB8 for which we have recently investigate the function during the development of the spinal cord and identified transcriptional target genes in this structure (Wilmerding et al., 2021a).” in the introduction.

  2. It is not clear what the authors are pointing out in Fig 1 A. Fig A and B can be combined as in Fig 1 C.

Fig 1A shows transfected cells in dorsal view of electroporated embryos three days post-electroporation to help readers that are not familiar with electroporation to apprehend where is the area of the embryo targeted. As suggested by reviewer 2, in the revised version we combined Fig A and B in Fig1A and added a line in the picture where whole mount embryos are shown to represent transverse section at the trunk level.

Images in Fig 1 B and subsequent figures, show less transfected cells in the MEK1ca+HOXB8 condition.

We agree with reviewer 2, there is less GFP+ cells in MEK1ca+HOXB8 condition than in other ones. Transfection by electroporation is not a 100% reproducible technique and it might have some variation between embryos in general. We however believe that the reason why there is less MEK1ca+HOXB8 cells than other conditions is because the GFP+ cell in this condition are more likely to die than in other conditions, coherent with the CASP3+ quantification made one day post-electroporation (Fig 1C-D).

 3. Also, both MEK1ca and HOXB8 are labelled with GFP and it’s not clear how much of MEK1ca and HOXB8 is expressed in the MEK1ca+HOXB8 condition.

We agree with reviewer 2 that we should have specified in the text of the submitted version that in the trunk neural tube of the chicken embryo, co-transfection is nearly total, as assed by co-electroporation of vector expressing different reporter protein as for example in our recent published study where the reporters are GFP and RFP (Wilmerding et al., 2021, Development, doi:10.1242/dev.195404). We add this precision in the revised version of the manuscript in the materials and methods section.

  4. In Fig 1 C image, looks like MEK1ca shows more CASP3 (red) staining than MEK1ca+HOXB8 condition and clearly does not co relate with the quantification in Fig 1D.

We apologize for not understanding this comment from reviewer 2. We believe we have instead chosen an image that matches the quantification in Fig1D.

You can do a western blot for casp3 for the different conditions.

For western blot, we agree that it would be an alternative way to quantify apoptosis, but this would have required microdissecting the neural tube, sorting the transfected cells by FACS and pooling a large number of embryos as we did for the RNAseq (so a much more complicated protocol), and we don't think this would necessarily provide more reliable data than immunohistochemistry. We hope the reviewer will understand our point of view.

      Also, does inhibition of CASP3 in the MEK1ca+HOXB8 condition revert the tumor suppressor effect of HOXB8?

We found the question addressed by the reviewer 2 “if inhibition of apoptosis in the MEK1ca+HOXB8 condition revert the tumor suppressor effect of HOXB8” very pertinent. This would have required, for example, co-electroporation of MEK1ca+HOXB8 with a vector expressing the P35 protein, known to inhibit apoptosis (Sahdev et al., 2010, doi: 10.1002/btpr.339). Since HOX8 reverses the effect of MEK1ca to a very large extent at the transcriptomic level, it seems unlikely that inhibiting death would be sufficient to suppress the tumour suppressor effect of HOXB8 but it would be interesting in a future study to test this hypothesis.

  5. The authors also do not show a pCIG-HOXB8 vector alone condition in Fig 1, although they mention HOXB8 is endogenously expressed in the trunk neural tube.

pCIG-HOXB8 vector alone condition is not show in Fig1 because HOXB8 gain of function is the focus of an article we published recently, in which we show in detail the phenotype of HOXB8 gain of function alone (Wilmerding et al., 2021, Development, doi:10.1242/dev.195404). We added in the results section of the revised version of the manuscript : “We recently analyzed the function and identified target genes of HOXB8 in the developing spinal cord of chicken embryo (Wilmerding et al., 2021a). In particular, we demonstrated that the gain of function of HOXB8 in the trunk neural tube of 2-days old chicken embryo done by electroporation of the pCIG-HOXB8 expressing vector (co-expressing nuclear GFP) increases cell death and the amount of the transcript of the gene coding for the tumor suppressor LZTS1 (Wilmerding et al., 2021a)”, at the beginning of the first paragraph.

    6. Fig 2D legend for the heat map mentions lines 1 2 and 3 and the Fig shows only 2 conditions and is confusing. I would suggest including genes upregulated by MEK1ca alone

We apologize for this error and thank reviewer 2 for having mentioned it. We made the correction in the revised version of the manuscript, and we changed the corresponding legend figure. The analysis of the genes upregulated alone by MEK1ca was the focus of another study (Wilmerding, A., et al 2021, Preprint bioRxiv doi: 10.1101/2021.06.10.447891), and this heatmap is already present in that manuscript.

Minor comments

  1. Lines 287 and 330- correct all percentages to 42.3 and 30.4 instead of 42,3 and 30,4 Thank you for this comment
  2. Needs to be re written. Many errors, for example line 300- Among these genes is as example…..line 267- To further the analysis of the RNA-seq to identify molecular mechanisms implicated…line 453- Suggest several non-exclusive mechanisms though which HOXB8 acts as tumor…

   Thank you for corrections

    3. Include quantification for all the immunofluorescent images

The only two immunofluorescences in the figures are Fig1F and Supplementary Figure1. We quantified Fig1F in Fig1G, but we do not think that quantification for Supplementary Figure 1 is necessary because it is F-ACTIN staining used to highlight tissue organization. 

     4. Fig 4A mentions WT chicken embryo, what is the context of using the term wild type here?

WT refer to non electroporated Wild Type chicken embryo.

   5. Fig 3 B, mention the genes in cluster 3, 5 and 8?

All genes of the 12 clusters are listed in table 5. We mention the genes from the clusters that we think are most relevant.

   6. Line 540 and 551- Italicize

    7. Line 554- 45mn.. 45 min?

     8. Line 555- …They were treated X minutes with proteinase K 10μg/ml at RT. X minutes?

    9. Keep units uniform.. either h or hour, min or minutes

Thank you for this comments, we have changed 45mn in 45 minutes in the text and remove X min.

    10. Supplementary fig 6 – Is this image with MEK1ca alone?

This is a wild type (WT) embryo. We have added the precision in the Figure 6 of the revised manuscript.

    11. Final concentration of DNA delivered to embryo for electroporation is mentioned as between 1 to 2μg/μl. Mention the exact concentrations and volume injected in the neural tube lumen.

We apologized for our lack of precision. We electroporated at a final concentration of 2μg/μl and the concentration of each vector was detailed in the text. The volume injected in the neural tube lumen is about 0.5µl.  Because the neural tube lumen is not a close cavity, as at HH12 both extremities are still not close, any excess volume injected will diffuse.